# Counteracting Effects of Glutathione on the Glutamate-Driven Excitation/Inhibition Imbalance in First-Episode Schizophrenia: A 7T MRS and Dynamic Causal Modeling Study

**DOI:** 10.3390/antiox10010075

**Published:** 2021-01-08

**Authors:** Roberto Limongi, Peter Jeon, Jean Théberge, Lena Palaniyappan

**Affiliations:** 1Robarts Research Institute, University of Western Ontario, London, ON N6A 3K7, Canada; lpalaniy@uwo.ca; 2Department of Medical Biophysics, University of Western Ontario, London, ON N6A 3K7, Canada; yjeon4@uwo.ca (P.J.); jtheberge@lawsonimaging.ca (J.T.); 3Imaging Division, Lawson Health Research Institute, London, ON N6C 2R5, Canada; 4Department of Psychiatry, Schulich School of Medicine and Dentistry, University of Western Ontario, London, ON N6A 3K7, Canada; 5Diagnostic Imaging, St. Joseph’s Health Care, London, ON N6A 1Y6, Canada

**Keywords:** dynamic causal modeling, glutamate hypofunction, salience network, glutathione, schizophrenia

## Abstract

Oxidative stress plays a key role in the pathophysiology of schizophrenia. While free radicals produced by glutamatergic excess and oxidative metabolism have damaging effects on brain tissue, antioxidants such as glutathione (GSH) counteract these effects. The interaction between glutamate (GLU) and GSH is centered on N-Methyl-D-aspartate (NMDA) receptors. GSH levels increase during glutamate-mediated excitatory neuronal activity, which serves as a checkpoint to protect neurons from oxidative damage and reduce excitatory overdrive. We studied the possible influence of GSH on the glutamate-mediated dysconnectivity in 19 first-episode schizophrenia (FES) patients and 20 healthy control (HC) subjects. Using ultra-high field (7 Tesla) magnetic resonance spectroscopy (MRS) and resting state functional magnetic resonance imaging (fMRI), we measured GSH and GLU levels in the dorsal anterior cingulate cortex (dACC) and blood-oxygenation level-dependent activity in both the dACC and the anterior insula (AI). Using spectral dynamic causal modeling, we found that when compared to HCs, in FES patients inhibitory activity within the dACC decreased with GLU levels whereas inhibitory activity in both the dACC and AI increased with GSH levels. Our model explains how higher levels of GSH can reverse the downstream pathophysiological effects of a hyperglutamatergic state in FES. This provides an initial insight into the possible mechanistic effect of antioxidant system on the excitatory overdrive in the salience network (dACC-AI).

## 1. Introduction

The role of oxidative stress in molecular mechanisms of neurodegenerative diseases has been investigated thoroughly [1]; however, the details of its involvement in pathophysiology of schizophrenia are not completely understood [2]. While free radicals produced by glutamatergic excess and oxidative metabolism have damaging effects on brain tissue, antioxidants such as glutathione (GSH) counteract the “toxic effect” of oxidative stress [3]. This antioxidant protection is reportedly aberrant in schizophrenia, with some patients having a notable reduction of GSH in the brain [4,5,6] and others with relatively better outcomes having higher than expected levels [7]. Several pharmacological approaches to improve glutathione-mediated antioxidant capacity are currently being studied in schizophrenia [8].

The interaction between glutamate (GLU) and glutathione is centered in the N-Methyl-D-aspartate (NMDA) receptors as well as the neuro-glial metabolic shuttling. GSH levels increase in response to glutamate-mediated excitatory neuronal activity. This increase is mediated by the NMDA receptor system [9] as well as by the conversion of glutamate to GSH in glial cells [10]. GSH has also been reported to have a direct signaling effect, by facilitating NMDA function [11,12] and increasing the inhibitory tone of microcircuits [13]. In humans, correlations between GSH and GLU levels in the dorsal anterior cingulate cortex (dACC) and the anterior insula (AI) have been reported [7]. Taken together, a glutamate-mediated GSH increase serves as a checkpoint to protect neurons from oxidative damage and reduce excitatory overdrive.

GSH depletion in the developing brain affects the interneurons that provide the NMDA-mediated inhibitory checkpoint for glutamatergic activity. This early NMDA hypofunction is considered to disrupt the normal excitation–inhibition balance and prime cortical networks for glutamatergic excess and excitatory overdrive in schizophrenia [7]. At present, it is unclear if higher levels of GSH can overcome this excitatory overdrive by restoring the inhibitory tone in patients with schizophrenia.

We have recently provided the first imaging evidence for the NMDA hypofunction model by demonstrating that glutamate levels are indeed related to a reduced inhibitory tone in schizophrenia [14]. We studied glutamate levels from the dorsal anterior cingulate cortex (dACC) and studied the salience network that connects dACC with the anterior insula (AI), using a biologically realistic neural model of resting-state functional magnetic resonance imaging (fMRI) data. In the current work, we extended this observation to study if GSH influences the glutamate-mediated dysconnectivity in first-episode schizophrenia (FES). Such an influence, if demonstrated, will add credibility to the notion that GSH can physiologically counteract the glutamate-mediated excitation–inhibition imbalance in schizophrenia.

## 2. Materials and Methods

### 2.1. Participants

We recruited 39 subjects in total; 19 with FES and 20 healthy control (HC) subjects (Table 1). This patient sample has been previously reported [14]. FES was defined as (1) patients with first clinical presentation with psychosis, (2) symptoms satisfying the [Diagnostic and Statistical Manual of Mental Disorders, Fifth Edition, 16] criteria A for schizophrenia, and (3) patients with less than 2 weeks of lifetime antipsychotic exposure. By relying on the best estimate procedure, as described in Leckman, et al. [15], and the Structured Clinical Interview for DSM-5, every FES patient received a consensus diagnosis from 3 psychiatrists after approximately 6 months. Each patient satisfied the DSM-5 criteria for schizophrenia spectrum disorders. Specifically, 15 patients satisfied the criteria for schizophrenia and 3 patients satisfied the criteria for schizoaffective disorder. Clinical data at 6 months were not available from 1 patient. However, the available baseline data suggested a diagnosis of schizophreniform disorder. Based on the above, we used the term FES to describe the patient group—capturing all the schizophrenia spectrum disorders. For these patients, we computed the defined daily dose (DDD), and the sample’s DDD mean was 1.05. This suggested that, on average, a patient had had 1-day worth of exposure to s standardized dose (at the time of assessment) when scanned. At the time of scan, approximately 40% of patients had not been exposed to any antipsychotic. Therefore, this sample can be considered as an acutely unwell, untreated, first-episode sample of schizophrenia spectrum disorders. Participants were recruited continuously from the Prevention and Early Intervention Program for Psychosis in London, Ontario. Finally, we assessed symptoms using the eight-item version of the positive and negative syndrome scale [16] (Table 1).

### 2.2. Magnetic Resonance Spectroscopy (MRS) Acquisition and Analysis

All data was acquired using a 680-mm neuro-optimized 7 T MRI scanner (Siemens MAGNETOM Plus, Erlangen, Germany) equipped with an AC84 II head gradient coil and an 8-channel Tx, 32-channel Rx radiofrequency coil. We defined a 2.0 × 2.0 × 2.0 cm (8 cm^3^) ^1^H-MRS voxel on the bilateral dACC (Figure 1). To this aim, we used a two-dimensional sagittal anatomical image (37 slices, TR = 8000 ms, TE = 70 ms, flip-angle (*α*) = 120°, thickness = 3.5 mm, field of view = 240 × 191 mm) as reference. We defined the voxel position both by setting the posterior face of the voxel in coincidence with the precentral gyrus and by setting the position of the inferior face of the voxel to the most caudal point not part of the corpus callosum. We set the voxel angle tangentially to the corpus callosum. A semi-LASER ^1^H-MRS sequence (TR = 7500 ms, TE = 100 ms, bandwidth = 6000 Hz, N_avg_ = 2048) was used to acquire 32 channel-combined, [17] VAPOR water-suppressed spectra as well as a water-unsuppressed spectrum (N_avg_ = 1) to be used for spectral editing and quantification. We asked all participants to fix their gaze on a white cross (50% gray background) during MRS acquisition. All scanning took place at the Centre for Functional and Metabolic Mapping of Western University, London, Ontario.

Based on Near and colleagues [18], we phase- and frequency-corrected the 32 spectra. Following, we computed a single average spectrum which was used in all subsequent analyses. Spectrum’s line shape deconvolution and removal of a residual water signal was performed via QUECC [19] (a combination of quantification improvement by converting lineshapes to the Lorentzian type, QUALITY, and eddy current correction, ECC) and Hankel singular value decomposition (HSVD) [20], respectively. Spectral fitting was done via fitMAN [21] (a time-domain fitting algorithm that uses a non-linear, iterative Levenberg-Marquardt minimization algorithm and echo-time, field strength, and pulse sequence specific prior knowledge templates). The metabolite-fitting template included 17 brain metabolites including glutamate and glutathione reported here. The other metabolites were N-acetyl aspartate, N-acetyl aspartyl glutamate, alanine, aspartate, choline, creatine, γ-aminobutyric acid (GABA), glucose, glutamine, glycine, lactate, myo-inositol, phosphorylethanolamine, scyllo-inositol, and taurine. Due to the long echo time used, no significant macromolecular contribution was expected. Metabolite quantification was then performed using Barstool [22] with corrections made for tissue-specific (gray matter, white matter, CSF) T_1_ and T_2_ relaxation through partial volume segmentation calculations of voxels mapped onto T_1_-weighted images acquired using a 0.75-mm isotropic MP2RAGE sequence (TR = 6000 ms, TI_1_ = 800 ms, TI_2_ = 2700 ms, flip-angle 1 (*α*_1_) = 4°, flip-angle 2 (*α*_2_) = 5°, FOV = 350 × 263 × 350 mm, T_acq_ = 9 min 38 s, iPAT_PE_ = 3 and 6/8 partial k-space). All spectral fits underwent visual quality inspection as well as Cramer–Rao lower bounds (CRLB) assessment for each metabolite.

The quality of metabolite quantification was measured using CRLB percentages for both groups using a CRLB threshold < 30% for glutathione to determine inclusion toward further analyses, in line with our prior study [23]. There was no significant difference in CRLB between the FES patients and HC subjects for both metabolites being reported in this study. A sample of fitted spectrum for a single participant is presented in Figure 1.

### 2.3. Bayesian Analysis

We estimated the posterior distribution of the (estimated) between-group differences in GLU and GSH by means of the generalized linear model within the context of hierarchical the Bayesian parameter estimation as follows:(1)$$Metabolite_{i}=\beta_{0}+\underset{group}{\sum }\beta_{group}x_{group}\left(i\right)$$
where the data conformed to a normal distribution around the predicted value (metabolite concentration) with a (wide) data-scaled uniform prior distribution for the standard deviation (σ_i_). The baseline parameter ($\beta_{0})$ had a data-scaled normal prior distribution with mean equal to the data mean and (wide) standard deviation relative to the standard deviation (SD_data_) of the data (1/(SD_data_ × 5)^2^). Group deflection parameters ($\beta_{group})$ had normal prior distributions with mean zero and a Gamma prior distribution for the standard deviation σ_β_ with data-scaled shape and rate parameters (SD_data_/2 and 2 × SD_data_ respectively). This meant that σ_β_ provided informed priors on each group’s (deflection) parameter. In other words, groups would act as priors between each other. In total, we estimated posterior distributions of five free parameters (σ_i_, $\beta_{0}$, $\beta_{HC}$, $\beta_{FES}$, and σ_β_). Posteriors were estimated in the R-software equivalent of “just another Gibbs sampler” (RJAGS) [24] using Markov chain Monte Carlo methods, drawing 11,000 samples (thinning = 10). We reported the proportion of the posterior distribution (i.e., posterior proportion, PP) of the between-groups difference in GSH and GLU levels along with the 95% highest density interval (HDI) of the posterior proportion. The posterior distributions and HDIs of the relevant effect sizes were also reported.

### 2.4. Resting-State fMRI

Resting-state whole-brain functional images were acquired over 6 min (360 volumes in total). We used a gradient echo planar imaging (EPI) sequence (TE = 20 ms, TR = 1000 ms, flip angle = 30 deg, field of view = 208 mm, voxel dimension = 2 mm isotropic in 63 contiguous slices). EPI data acquisition was accelerated using GRAPPA = 3 and a multi-band factor = 3. A 3D, T1-weighted MP2RAGE anatomical volume (TE/TR = 2.83/6000 ms, TI1/TI2 = 800/2700 ms) at 750 µm isotropic resolution was acquired as an anatomical reference.

### 2.5. Spectral Dynamic Causal Modeling of Network Connectivity

We fit spectral dynamic causal models to the fMRI time series data to quantitatively infer how the fMRI timeseries were generated by (unobserved) neural activity of coupled neuronal populations between the dACC and the AI during resting state [25]. At present, dynamic causal modeling is considered the most physiologically grounded technique to infer the effective connectivity between brain regions [26]. Specifically, a spectral dynamic causal model is a special case of “generative models” in which the neural causes are hidden states and the blood-oxygenation level-dependent (BOLD) signals are observed measurements. Therefore, the (generative) dynamic causal model comprised one evolution function (2) where x’(t) is the rate of change of the neuronal states x(t), θ represents the unknown parameters of the effective connectivity, and v(t) represents the states noise. The output of the evolution function, x(t), was mapped onto an observed function (3) where y(t) is the measured BOLD signal, φ represents the unknown parameters, and e(t) is the observation noise. Crucially, the diffusion (or noise) terms in (2) and (3) could be parameterized. Therefore, the evolution function became a random differential equation. For a thorough mathematical description of the generative model we refer the interested reader to both Friston, et al. [27] and Razi, Kahan, Rees and Friston [25].
(2)$$x^{\prime }\left(t\right)\;=f\left(x\left(t\right),\;\theta \right)+\;v\left(t\right)$$
(3)$$y\left(t\right)\;=h\left(x^{\prime }\left(t\right),\varphi \right)+\;e\left(t\right)$$

Our dynamic causal model of the dACC-AI network represented both intrinsic or within-region (GABAergic) connections and extrinsic (between-region) glutamatergic neuronal populations within each region [28,29]. Each population comprised self-inhibition connections (which are fixed parameters). Two free parameters were fit to the fMRI data: Interregional excitatory-to-excitatory connections and within-region inhibitory-to-excitatory connections (Figure 2). Each of these parameters was the log of a scaling factor, which was multiplied by the default connection strength: 1/8Hz for between-region connections and −1/8Hz for within-region connections. This formulation enforces positivity or negativity constraints on the connections, and gave the parameters a simple interpretation, as follows. Between-region connections were excitatory, so more positive values corresponded to greater excitation and more negative values corresponded to less excitation. Conversely, positive values of inhibitory connections indicated greater inhibition and less positive values indicated less inhibition.

At a subject level, the analysis was the same as we have previously reported [15]. Specifically, we estimated the resting-state effective connectivity within the dACC-AI network by fitting a fully connected model [25,29]. To this aim, realignment, normalization (to MNI space), and spatial smoothing (4-mm full width at half maximum with a Gaussian Kernel) were performed on the functional images. A general linear model (including six head movement parameters and time series corresponding to the white matter and cerebrospinal fluid as regressors) was fit to the images. A cosine basis set with frequencies ranging from 0.0078 to 0.1 Hz was also included in the general linear model [30]. Images were high-pass filtered to remove slow frequency drifts (<0.0078 Hz). By using an F contrast, we identified regions with blood oxygen level fluctuations within frequencies ranging from 0.0078 to 0.1 Hz [30]. Time series that summarized the activity within spheres (8-mm radius) in the right AI (MNI coordinates X = 38, Y = 20, Z = −4) and in the right DACC (MNI coordinates *x* = 1, y = 16, z = 38) were extracted and used to specify the dynamic causal models.

At a group level, we relied on parametric empirical Bayes (PEB) [31,32,33] to estimate the effect of GLU and GSH on connectivity parameters. We estimated a “two-metabolite” model with which we aimed to evaluate the evidence in support of the hypothesis that both GLU and GSH best explained the effective connectivity within the two-node network. The design matrix of the two-metabolite model comprised one column coding for group membership, one column comprising the mean-centered GLU levels, one column comprising the GLU × group interaction, one column comprising the mean-centered GSH levels, and one column comprising the GSH × group interaction (in this order). The design matrix also comprised a constant (column of ones). We compared the evidence in support of this model against the evidence in support of an “only-group” model, a reduced model comprising only the effect of the group on connectivity parameters.

We adjudicated between the two-metabolite and the only-group models by means of Bayesian model selection [34]. Specifically, we evaluated the evidence of each model (as estimated by the negative variational free energy, F). In principle, the strongest evidence is ascribed to the model with the least negative free energy. However, it is useful to assess the evidence of a given model relative to the evidence ascribed to other models. This is achieved by means of Bayesian model comparison in which the evidence of a given model (F1) is compared to the evidence of the model with the most negative free energy (F2), yielding the log of the Bayes factor (lnBF_1_ = F_1_−F). In terms of posterior probability (PP), a BF > 20 is equivalent to a PP > 0.95 [34] which indexes very strong evidence. Therefore, we relied on a PP > 0.95 as a decision rule (i.e., threshold) for model selection [35]. Finally, the sum of the posteriors of all models’ posteriors equaled to 1.

## 3. Results

### 3.1. Between-Group Comparison in Metabolite Levels

The Bayesian linear model revealed higher GSH levels in the FES group than in the HC group (mode of the between-groups difference = 0.25, PP = 0.98; mode of the effect size = 0.71, PP = 0.98). The Bayesian analysis did not reveal an effect of the group on GLU levels (mode of the between-groups difference = 0.17, PP = 0.84; mode of the effect size = 0.1, PP = 0.84). Summary statistics of the posterior distributions of the model’s parameters are reported in Table 2, and Figure 3 shows the posterior distributions of the estimated between-group difference in GSH and GLU levels.

### 3.2. Spectral Dynamic Causal Models of Effective Connectivity

The two-metabolite model, comprising the effect of GSH and GLU, performed better than the group-only model (PP > 0.99). As shown in Figure 4 and in line with our previous work [15], the activity of the inhibitory neurons in the dACC decreased as a function of GLU in the FES group (PP > 0.95). Crucially, in the current model, this effect was reversed by GSH, which was associated with increased inhibitory activity. This effect of GSH on IE connections (see also Figure 2 for reference) was observed not only in the dACC (PP > 0.95) but also in the inhibitory neural population of the AI (PP > 0.95).

## 4. Discussion

In drug-naïve patients with first episode psychosis, GSH levels were higher than in HC subjects. Higher GSH levels were related to stronger intrinsic inhibition within dACC and AI nodes of the salience network, a large-scale network known to play a cardinal role in schizophrenia symptoms [36]. This effect was in direct contrast to the relationship between higher GLU levels and putative disinhibition (i.e., reduced intrinsic inhibition) within the dACC. Our model provided an explanation for how higher levels of GSH can reverse the downstream pathophysiological effects of a putative hyperglutamatergic state in FES.

The presence of higher levels of GSH in patients compared to controls was in contrast to our meta-analytic observation of a small reduction of GSH in schizophrenia [37]. Nevertheless, as we reported in the same meta-analysis, patients with bipolar disorder had a small increase in GSH levels compared to healthy controls, leading to the speculation that GSH levels may mark the outcomes of psychotic disorders rather than the diagnosis per se. In fact, our prior observation from an overlapping sample with longitudinal clinical data supports this idea [23]. Higher levels of GSH are likely to indicate a more favorable prognosis, with a quicker response to antipsychotics in first-episode psychosis [23]. As such, the current sample of first-episode patients likely comprised subjects with more favorable outcomes than the chronic schizophrenia samples studied in our prior meta-analysis.

Our results also indicated that in early phases of psychosis, GSH may operate to reverse glutamate-mediated dysconnectivity. Specifically, stronger disinhibition of GABA neurons with higher levels of GLU reflected a hyperglutamatergic state in FES subjects—as indexed by the negative value of the parameter estimate representing inhibitory connections within the dACC. However, positive values of the effect of GSH indicated a direct relationship between the GSH level and a much stronger inhibitory (i.e., GABAergic) activity within both the dACC and AI. Since, as per the model’s assumptions, the net activity of a given neuronal population is a linear function of the relevant parameter estimates, the magnitudes of these estimates suggest that the hyperglutamatergic state is compensated (or restrained) by an “antioxidative” state. This is important especially because our FES subjects were untreated when these data were collected. On this basis, we speculate that a targeted increase in dACC GSH levels via antioxidant supplementation or targeting the Nrf2 pathway could improve patients’ response to antipsychotics [38]. More speculatively, this may assist in achieving an adequate response at lower-than-usual doses and cut down the total duration of higher dose exposure, both of which are now argued by some as key strategies to improve functional recovery in psychosis [39]. In the context of antioxidant trials, this also speaks to a stratification strategy based on baseline levels of GSH—as suggested in previous works [40].

The robustness of our results rests on several methodological strengths. First, we used 7T-MRS sequence with improved specificity to detect GSH resonance with reduce macromolecular interference [41]. This level of specificity contrasts with the specificity achieved using 3T-MRS [42,43]. Furthermore, the effective connectivity model is biologically grounded despite the fact that it does not consider the variability of inhibitory neuronal populations that have been recently reported [44]. Regardless of this limitation, the two-neuronal-population model was enough to evaluate our hypothesis. Finally, from our cross-sectional data, we cannot infer if the GSH increase is secondary to increased intrinsic inhibition or vice versa. Longitudinal fMRI and MRS data on GLU and GSH could potentially address this limitation in the future.

## 5. Conclusions

In summary, our data and computational model provide initial clues to understand the mechanistic effect of GSH on the previously reported hyperglutamatergic state within the dACC–AI network. As summarized in Figure 5, redox imbalance in early life may prime the brain for excitatory overdrive in schizophrenia; but if an appropriate increase in GSH accompanies glutamatergic excess, the inhibitory tone may be strengthened in compensation.

## Figures and Tables

**Figure 1 antioxidants-10-00075-f001:**
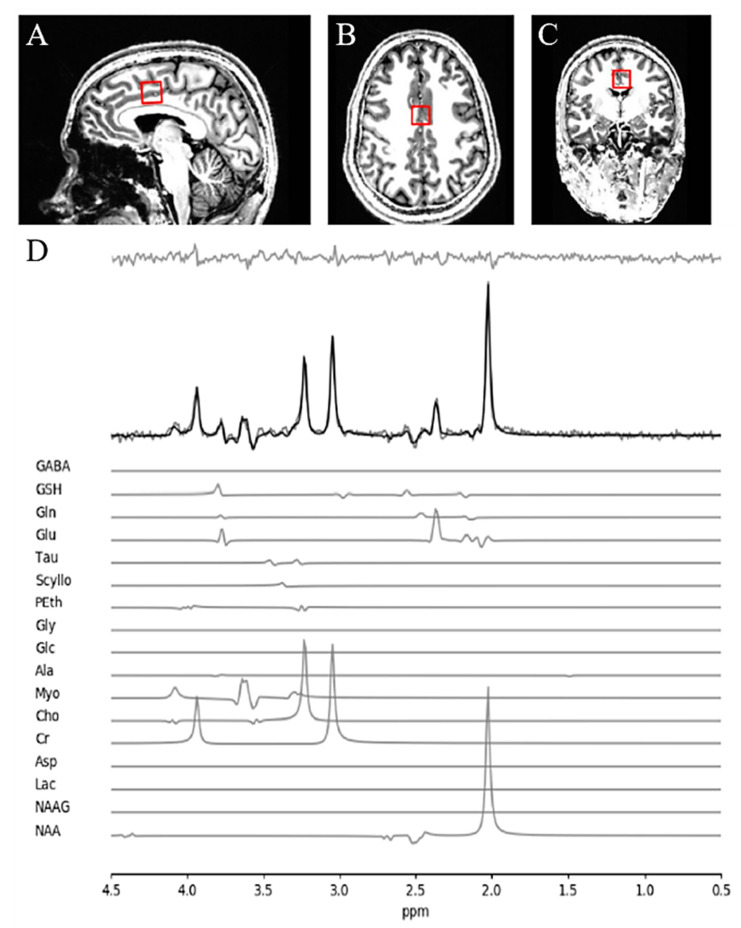
MRS voxel and spectra. (**A**) Sagittal, (**B**) axial, and (**C**) coronal view of voxel positioning on the dorsal anterior cingulate cortex (dACC). (**D**) Sample spectra obtained from a single healthy participant. The bold black line represents the fitted spectra with the residuals above (the gray line above the fitted curve) and each individual metabolite contribution below.

**Figure 2 antioxidants-10-00075-f002:**
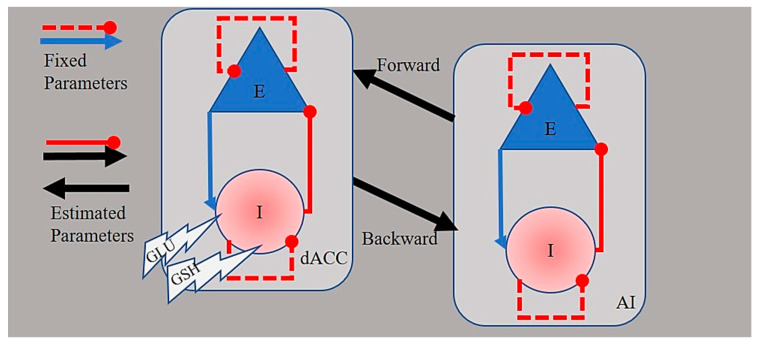
Circuit model of the dACC–AI network. In a two-state dynamic causal model, both inhibitory (I) and excitatory (E) neuronal populations comprise self-inhibitory connections (dashed red lines with oval arrows) which are fixed parameters. Excitatory intrinsic connections (i.e., from E neurons to I neurons, EI, solid lines with blue arrows) activate I neurons, and are fixed parameters. Inhibitory intrinsic (GABAergic) connections (i.e., from I neurons to E neurons, IE, solid lines with oval red arrows) inhibit E neurons and are free parameters. Extrinsic forward and backward (glutamatergic) connections (black arrows) are also free parameters.

**Figure 3 antioxidants-10-00075-f003:**
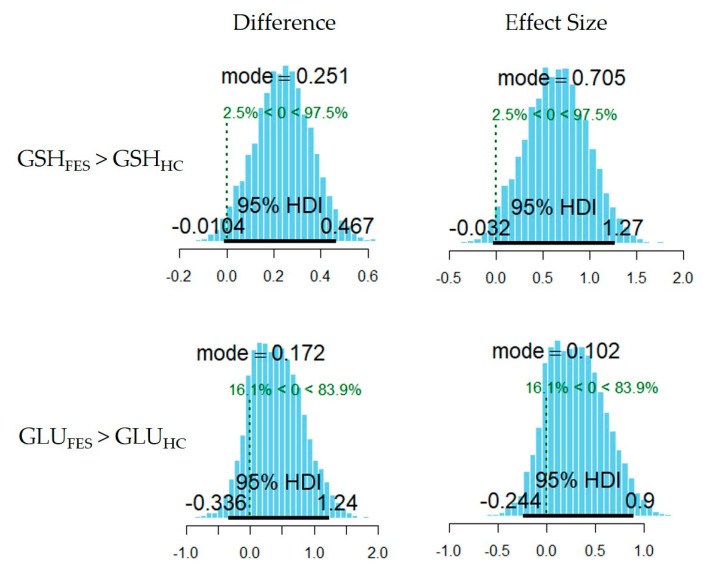
Posterior distributions of the estimated between-group differences and effect sizes in GLU (lower panels) and GSH (upper panels) levels.

**Figure 4 antioxidants-10-00075-f004:**
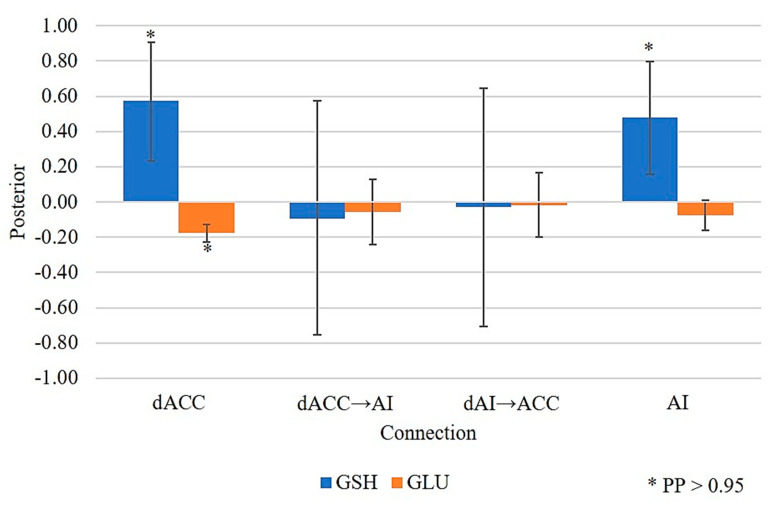
Counteracting effect of the GSH on the hyperglutamatergic state in the dACC–AI network. As previously reported [15], in the dACC the effect of GLU on IE connections (see also Figure 2 for reference) was weaker in FES patients than in HC subjects (indexed by the negative parameter estimate). This indicates stronger disinhibition of excitatory neuronal population with higher levels of GLU in patients than in controls, leading to a hyperglutamatergic state. Crucially, positive parameter estimates in blue indicate the effect of GSH on inhibitory (i.e., GABAergic) activity within the dACC and AI, this effect being stronger in FES patients than in HC subjects. Since the net activity of a given neuronal population is a linear function of the relevant parameter estimates, the magnitudes of these estimates indicate that the glutamatergic influence on intrinsic connectivity in the dACC is compensated by the “antioxidative” state.

**Figure 5 antioxidants-10-00075-f005:**
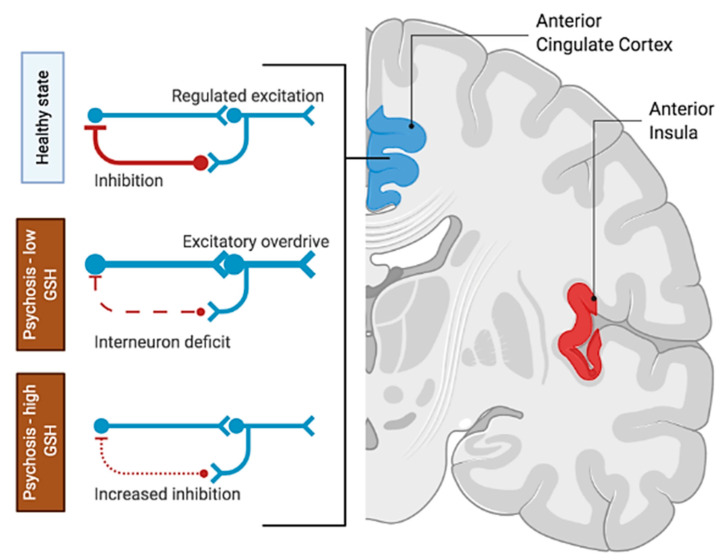
Summary of the counteracting effect of GSH on redox imbalance in the salience network. In a healthy state, the excitation–inhibition balance is achieved via the interplay of excitatory and feedback inhibitory neuronal populations. Glutamate-mediated dysconnectivity in psychosis (likely caused by GSH depletion in the developing brain) would manifest in terms of interneuron deficit, leading to excitatory overdrive (i.e., hyperglutamatergic state). The resulting excitatory overdrive can be reduced by increasing inhibition via a compensatory increase in the levels of GSH, at least in a subset of patients. This illustration was produced using biorender.com

**Table 1 antioxidants-10-00075-t001:** Demographics and clinical characteristics.

Subject	Group	SOFAS	Parental SES (NSSEC)	Age at Study Date	Gender	PANSS Sub-Scores							
						P1	P2	P3	N1	N4	N6	G5	G9
1	HC	73	3	26	Male	-	-	-	-	-	-	-	-
2		85	4	23	Female	-	-	-	-	-	-	-	-
3		79	2	17	Male	-	-	-	-	-	-	-	-
4		85	5	23	Male	-	-	-	-	-	-	-	-
5		87	1	16	Female	-	-	-	-	-	-	-	-
6		81	5	25	Male	-	-	-	-	-	-	-	-
7		83	2	16	Male	-	-	-	-	-	-	-	-
8		86	2	16	Male	-	-	-	-	-	-	-	-
9		80	3	29	Male	-	-	-	-	-	-	-	-
10		79	2	22	Female	-	-	-	-	-	-	-	-
11		79	2	23	Female	-	-	-	-	-	-	-	-
12		83	1	20	Male	-	-	-	-	-	-	-	-
13		80	4	20	Male	-	-	-	-	-	-	-	-
14		80	2	20	Male	-	-	-	-	-	-	-	-
15		85	3	20	Male	-	-	-	-	-	-	-	-
16		80	2	20	Female	-	-	-	-	-	-	-	-
17		85	5	27	Female	-	-	-	-	-	-	-	-
18		85	5	22	Female	-	-	-	-	-	-	-	-
19		80	5	18	Female	-	-	-	-	-	-	-	-
20		85	2	22	Female	-	-	-	-	-	-	-	-
1	FES	40	4	19	Male	4	4	4	4	5	3	2	4
2		37	5	20	Male	5	4	5	2	3	1	3	4
3		40	2	19	Male	4	5	4	5	4	3	1	3
4		60	2	17	Male	5	1	5	3	4	3	1	4
5		30	4	18	Male	5	3	4	2	3	3	1	3
6		51	4	17	Female	5	1	5	3	5	3	3	2
7		34	5	24	Male	6	5	6	1	1	1	4	5
8		50	2	21	Male	5	4	4	2	4	2	1	4
9		25	2	25	Male	7	3	5	4	3	1	2	6
10		40	2	28	Male	6	4	2	1	1	1	1	3
11		33	2	20	Female	6	3	2	1	1	1	1	4
12		65	3	23	Female	4	2	5	3	1	2	1	3
13		25	2	23	Male	5	6	5	1	3	4	3	5
14		44	3	24	Female	5	4	2	1	1	1	1	5
15		20	2	23	Female	7	4	6	1	1	1	1	6
16		55	4	20	Male	5	1	5	4	3	1	1	3
17		50	1	27	Male	7	3	2	5	4	3	1	3
18		40	5	26	Female	5	1	5	1	5	1	1	3
19		45	4	19	Female	5	3	1	3	3	3	1	4

Note. HC = healthy control, FES = first episode schizophrenia, SOFAS = social and occupational functional assessment scale, NSSEC = national statistics socioeconomic status, delusions (P1), conceptual disorganization (P2), hallucinations (P3), blunted affect (N1), social withdrawal (N4), lack of spontaneity (N6), mannerisms (G5), unusual thoughts (G9).

**Table 2 antioxidants-10-00075-t002:** Parameter estimates (posteriors) of the hierarchical Bayesian linear model of the effect of group on glutamate (GLU) and glutathione (GSH) levels.

Metabolite	Parameter	Mean	Median	Mode	HDI (Low)	HDI (High)
GLU	β_0_	6.641	6.641	6.650	6.212	7.074
	β_FES_	0.203	0.195	0.086	−0.168	0.622
	β_HC_	−0.203	−0.195	−0.086	−0.622	0.168
	σ_β_	1.386	0.923	0.361	0.000	4.266
	σ_i_	1.374	1.358	1.309	1.076	1.727
GSH	β_0_	1.604	1.604	1.610	1.482	1.719
	β_FES_	0.118	0.117	0.118	−0.003	0.236
	β_HC_	−0.118	−0.117	−0.118	−0.236	0.003
	σ_β_	0.515	0.365	0.197	0.001	1.460
	σ_i_	0.374	0.369	0.362	0.294	0.469

Note. HDI highest density interval (95% of the most credible values), $\beta_{0}$ intercept, β_FES_ deflection parameter for the FES group, β_HC_ deflection parameter for the HC group, σ_β_ standard deviation of the baseline parameter, σ_i_ standard deviation of the predicted value.

## Data Availability

The data presented in this study are available on request from Dr. Lena Palaniyappan (lpalaniy@uwo.ca).

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
