# Peer review of "Counteracting Effects of Glutathione on the Glutamate-Driven Excitation/Inhibition Imbalance in First-Episode Schizophrenia: A 7T MRS and Dynamic Causal Modeling Study"

_antioxidants, 2021, doi:10.3390/antiox10010075_

Round 1

Reviewer 1 Report

The imbalance of oxidative stress has been subjected to a variety of neurodegenerative and psychological disorders. However, how does oxidative stress affects the pathogenesis of these neurological and psychological diseases is not fully understood. In this study, the authors used 7T MRS and resting state fMRI to measure GSH and GLU in the first episode schizophrenia (FES) patients to investigate the pathophysiological effects of a hyperglutamatergic state in FES. This is study is well conducted, the data is meaningful and clearly demonstrated the GSH levels regulate the dynamic of the FES. I only have minor comments.

  1. The figure is wrong labeled, there are two Figure 2. I cannot find Figure 1 in the context either.
  2. In line 211, the author indicated the parameters in Table 2 and 3. There is no Table 3.

Author Response

We thank the reviewer for the kind comments. As suggested, we addressed their minor comments.

1. The figure is wrong labeled, there are two Figure 2. I cannot find Figure 1 in the context either

Figure 1 is now labeled both in the relevant caption (line 138) and in the context (Lines 105 and 134).

2. In line 211, the author indicated the parameters in Table 2 and 3. There is no Table 3.

We now provide only one parameters table (Table 2, line 243)

Reviewer 2 Report

Consolidated Comments on “Counteracting Effect of Glutathione on the Glutamate-driven Excitation/Inhibition Imbalance in First-Episode Schizophrenia: A7T MRS and Dynamic Causal Modeling Study”

  1. The authors studied how an appropriate increase in GSH levels can potentially reversed the effects of a hyperglutamatergic state in FES in the dACC-AI network. To this aim, the authors measured the GSH and GLU levels via MRS and fMRI in control and FES patients, and then fitted these signals using dynamic causal modeling to identify the effect of GSH and GLU in the inhibitory activity within the dACC-AI network. Their results suggest that increased GSH can reverse the excitatory overdrive in FES patients by increasing inhibitory activity.
  2. The manuscript is overall well written, and the proposed methodology is appropriate. However, the following points need to be addressed:
    1. The Dynamic Causal Modeling (DCM) of Network Connectivity employed is described in the sub-section 2.5. However, this sub-section will significantly improve if the authors describe more thoroughly the system of differential equations used to obtain their fits with the different models tested at least as a supplementary material.
    2. The sentence in lines 63-65 needs some rewording, e.g.: “FES group included: (i) clinical presentation with psychotic symptoms and DSM-5 criteria [15]; (ii) less than 2 weeks of lifetime antipsychotic exposure.
    3. Figure 1 is mislabeled as Figure 2 in lines 85, 120 and 122.
    4. What does represent the gray line above the fitted curve in fig 2D (“Fig1”)? it is not defined in the figure legend.
    5. FES is misspelled in Table 1 and line 138 (one of the betas).
    6. Related to the point 2A: Add a legend to Figure 2-Line 172 to clearly explain the circuit model.
    7. Add a line sentence to briefly describe the analysis in [26] stated in line 173.
    8. In Figure 3 is not very clear from the figure itself nor from the figure legend, which distributions belongs to which group. Add labels to the distributions and a legend with a clear description of the figure.
    9. In Figure 4, the error bar in the dACC->AI bar seems wrong.
    10. The citation in line 249 is not numbered.
    11. Extra "provide" in line 258.
    12. What is the intension of the italic font in line 262?
    13. There is a font size change in line 287.
    14. Even though the cartoon in Figure 5 is almost self-explanatory, the authors should add a legend to the figure.

3. Other less critical points should be addressed:

  • Missing d, line 21 in ACC.
  • Extra comma in line 52.
  • Define all the abbreviations before using them in the text. E.g.: DSM in line 64; LP/KD in line 69.
  • The table titles (Table 1 & 2) would look better if place underneath each table, and the notes added as a legend in each table.
  • Extra dot in line 293.

Author Response

We thank the reviewer for the positive assessment. As detailed below, we now provide the appropriate modifications.

    1. The manuscript is overall well written, and the proposed methodology is appropriate. However, the following points need to be addressed:
  • The Dynamic Causal Modeling (DCM) of Network Connectivity employed is described in the sub-section 2.5. However, this sub-section will significantly improve if the authors describe more thoroughly the system of differential equations used to obtain their fits with the different models tested at least as a supplementary material.

Basically, a dynamic causal model is a special case of generative models with evolution and observed functions. For brevity and more precision, we rewrote this section and described the general forms of these two fundamental functions. Furthermore, we refer the interested reader to the relevant references where the system of differential equations is thoroughly described.

  • The sentence in lines 63-65 needs some rewording, e.g.: “FES group included: (i) clinical presentation with psychotic symptoms and DSM-5 criteria [15]; (ii) less than 2 weeks of lifetime antipsychotic exposure.”

We reworded this sentence as follows (now in lines 71-73): “Patients were included in the FES group if at first clinical presentation with psychotic symptoms they both met the DSM-5 [16] criteria A for schizophrenia and had less than 2 weeks of lifetime antipsychotic exposure. At the time of assesment, approximately 40% of patients had not been exposed to any antipsychotic.”

  • Figure 1 is mislabeled as Figure 2 in lines 85, 120 and 122.

We now labeled Figure 1 as such.

  • What does represent the gray line above the fitted curve in fig 2D (“Fig1”)? it is not defined in the figure legend.

The gray line represents the fitting residuals, it is now explicitly defined in the figure 1 legend.

  • FES is misspelled in Table 1 and line 138 (one of the betas).

FES spelling has now been corrected in Table 1.

  • Related to the point 2A: Add a legend to Figure 2-Line 172 to clearly explain the circuit model.

The circuit model is now detailed in the figure 2 legend.

  • Add a line sentence to briefly describe the analysis in [26] stated in line 173.

The analysis is now described.

  • In Figure 3 is not very clear from the figure itself nor from the figure legend, which distributions belongs to which group. Add labels to the distributions and a legend with a clear description of the figure.

Figure’s content and labels have been rearranged for more clarity.

  • In Figure 4, the error bar in the dACC->AI bar seems wrong.

The correct error bar has now been included in Figure 4.

  • The citation in line 249 is not numbered.

The citation is now numbered.

  • Extra "provide" in line 258.

“provide” has now been removed.

  • What is the intension of the italic font in line 262?

We now use normal font.

  • There is a font size change in line 287.

We corrected the font size.

  • Even though the cartoon in Figure 5 is almost self-explanatory, the authors should add a legend to the figure.

We added a descriptive legend to Figure 5.

  1. Other less critical points should be addressed:
  • Missing d, line 21 in ACC.

We now added ‘d’.

  • Extra comma in line 52.

Comma has been removed.

  • Define all the abbreviations before using them in the text. E.g.: DSM in line 64; LP/KD in line 69.

Abbreviations have been defined.

  • The table titles (Table 1 & 2) would look better if place underneath each table, and the notes added as a legend in each table.

Table titles are placed as per the journal requirement.

Extra dot in line 293.

  • The dot has been removed.

Reviewer 3 Report

Review of a manuscript “Counteracting Effect of Glutathione on the Glutamate-driven Excitation/ Inhibition Imbalance in First-Episode Schizophrenia: A 7T MRS and Dynamic Causal Modeling Study” by Roberto Limong and coauthors.  

Oxidative stress plays a key role in the pathology of many human diseases, including schizophrenia. Molecular mechanisms of oxidative stress are well studied in neurodegenerative diseases, and it is known that they are important in other diseases, including schizophrenia. However, the details of these interaction are poorly understood. The authors recently published the first imaging evidence for the NMDA hypofunction model and proved that the glutamate is related to reduced inhibitory tone in schizophrenia. In the submitted manuscript the authors present the results of glutathione effect on glutamate-mediated dysconnectivity in schizophrenia (FES). The results of these studies are important because they may demonstrate that glutathione can neutralize the glutamate-mediated excitation-inhibition imbalance in schizophrenia. The manuscript is well written, it contains new data which will be interesting for the readership of “Antioxidants”.

The following corrections should be made.

Abstract:

“We study if GSH influences the glutamate16 mediated dysconnectivity in 19 first episode schizophrenia (FES) and…” should be rewritten as “We study if GSH influences the glutamate mediated dysconnectivity in 19 first episode schizophrenia (FES) patients and…”

Introduction:

The authors should begin the Introduction with a more general description of oxidative stress role in brain diseases: ”The role of oxidative stress in molecular mechanisms of neurodegenerative diseases is investigated thoroughly [Ref 1: Emamzadeh et al. Parkinson’s disease: Biomarkers, Treatment, and Risk Factors. Front. Neurosci., 30 August 2018, 12:612], however, the details of its involvement in pathophysiology of schizophrenia is not completely understood [2 Flatov et al., 2013]

Materials and Methods. Lines 159-161

This effective connectivity is assumed to represent both intrinsic or within region (GABAergic) connections and extrinsic (between region) glutamatergic influences. Two-state DCM assumes excitatory and inhibitory populations of neurons within a region.

A reference (references) should be placed here.

Results

Figure 3. The text under the right upper panel is not seen clearly. It should be moved down and located at the same level as under the left upper panel.

Discussion

Line 259: ”Our results provide also indicate that in early phases of psychosis GSH may operate…” It’s unclear what the authors want to say by saying “provide also indicate”

Conclusion

Lines 286-287:” to understand the mechanistic effect of GSH in the previously reported aberrant hyperglutamatergic effect in the dACC287 AI network.” This is a clumsy sentence. It should be rewritten in a more clear way.

Author Response

We thank the reviewer for the comments and suggestions

The following corrections should be made.

Abstract:

“We study if GSH influences the glutamate16 mediated dysconnectivity in 19 first episode schizophrenia (FES) and…” should be rewritten as “We study if GSH influences the glutamate mediated dysconnectivity in 19 first episode schizophrenia (FES) patients and…”

We modified the abstract as suggested

Introduction:

The authors should begin the Introduction with a more general description of oxidative stress role in brain diseases: ”The role of oxidative stress in molecular mechanisms of neurodegenerative diseases is investigated thoroughly [Ref 1: Emamzadeh et al. Parkinson’s disease: Biomarkers, Treatment, and Risk Factors. Front. Neurosci., 30 August 2018, 12:612], however, the details of its involvement in pathophysiology of schizophrenia is not completely understood [2 Flatov et al., 2013]

We modified the intro as suggested and the new reference was included.

Materials and Methods. Lines 159-161

This effective connectivity is assumed to represent both intrinsic or within region (GABAergic) connections and extrinsic (between region) glutamatergic influences. Two-state DCM assumes excitatory and inhibitory populations of neurons within a region.

A reference (references) should be placed here.

This subsection was rewritten and the requested references were added.

Results

Figure 3. The text under the right upper panel is not seen clearly. It should be moved down and located at the same level as under the left upper panel.

Figure’s content and labels have been rearranged for more clarity.

Discussion

Line 259: ”Our results provide also indicate that in early phases of psychosis GSH may operate…” It’s unclear what the authors want to say by saying “provide also indicate”

This was a typo error, the word “provide” has been deleted.

Conclusion

Lines 286-287:” to understand the mechanistic effect of GSH in the previously reported aberrant hyperglutamatergic effect in the dACC287 AI network.” This is a clumsy sentence. It should be rewritten in a more clear way

We modified the sentence as requested.